# The Effect of Submaximal Exercise on Jugular Venous Pulse Assessed by a Wearable Cervical Plethysmography System

**DOI:** 10.3390/diagnostics12102407

**Published:** 2022-10-04

**Authors:** Erica Menegatti, Antonino Proto, Gianfranco Paternò, Giacomo Gadda, Sergio Gianesini, Andrea Raisi, Anselmo Pagani, Tommaso Piva, Valentina Zerbini, Gianni Mazzoni, Giovanni Grazzi, Angelo Taibi, Paolo Zamboni, Simona Mandini

**Affiliations:** 1Department of Environmental Science and Prevention, University of Ferrara, 44123 Ferrara, Italy; 2Department of Neuroscience and Rehabilitation, University of Ferrara, 44123 Ferrara, Italy; 3Department of Physics and Earth Sciences, University of Ferrara, 44122 Ferrara, Italy; 4Department of Translational Medicine, University of Ferrara, 44123 Ferrara, Italy; 5Department of Surgery, Uniformed Services University of the Health Sciences, Bethesda, MD 20814, USA; 6Healthy Living for Pandemic Event Protection (HL-PIVOT) Network, Chicago, IL 60612, USA

**Keywords:** jugular venous pulse, exercise, cardiac filling, cerebral venous drainage, wearable, cervical plethysmography, capacitive strain-gauge

## Abstract

The jugular venous pulse (JVP) is a one of the crucial parameters of efficient cardiovascular function. Nowadays, limited data are available regarding the response of JVP to exercise because of its complex and/or invasive assessment procedure. The aim of the present work is to test the feasibility of a non-invasive JVP plethysmography system to monitor different submaximal exercise condition. Twenty (20) healthy subjects (13M/7F mean age 25 ± 3, BMI 21 ± 2) underwent cervical strain-gauge plethysmography, acquired synchronously with the electrocardiogram, while they were carrying out different activities: stand supine, upright, and during the execution of aerobic exercise (2 km walking test) and leg-press machine exercise (submaximal 6 RM test). Peaks *a* and *x* of the JVP waveform were investigated since they reflect the volume of cardiac filling. To this aim, the Δ*ax* parameter was introduced, representing the amplitude differences between *a* and *x* peaks. Significant differences in the values of *a*, *x*, and Δ*ax* were found between static and exercise conditions (*p* < 0.0001, *p* < 0.0001, *p* < 0.0001), respectively. Particularly, the Δ*ax* value for the leg press was approximately three times higher than the supine, and during walking was even nine times higher. The exercise monitoring by means of the novel JVP plethysmography system is feasible during submaximal exercise, and it provides additional parameters on cardiac filling and cerebral venous drainage to the widely used heartbeat rate value.

## 1. Introduction

The jugular venous pulse (JVP) is a key parameter of efficient cardiovascular function, and it reflects pressure variation in the right atrium over the cardiac cycle. Consequently, the pulsation is reflected up to internal jugular veins (IJV), providing key information regarding cardiovascular function [1].

Although JVP has been used since the 1930s, unfortunately it can only be measured invasively by a venous catheter providing the central venous pressure (CVP). However, quite recently, it was assessed non-invasively by means of different devices, including ultrasound (US), photoplethysmography, and strain-gauge plethysmography [2,3,4,5,6]. In particular, the accuracy of US JVP was also confirmed versus CVP, paving the way to a widespread use of non-invasive JVP assessment [6,7]. 

The JVP waveform is composed by three ascents and three descents waives, which correspond to the pressure variation of the cardiac phases: atrial contraction (wave *a*) synchronized with P peak of ECG, followed by atrial relaxation (wave *x*) and tricuspid valve closure (wave *c*). After the QRS complex of the ECG the ventricular systole starts while passive atrial filling occurs followed by pressure drop (wave *x’*). Subsequently, after the T peak of ECG corresponding to ventricular repolarization, the maximum atrial filling will be obtained (wave *v*) before the tricuspid valve opening. This latter causes a sudden pressure decrease coincident with ventricular filling (wave *y*), and the cycle will start again. 

All the above-mentioned volumetric phases are synchronized with the electrical activity of the heart measurably with the electrocardiogram (ECG). Figure 1 shows the synchronization between the JVP and ECG signals.

Based on this, non-invasive JVP assessment was also used in special environments to monitor the cardiovascular function and cerebral venous drainage [8]. It was demonstrated that the reduction of cardiac output in cosmonauts during long term exposure to micro-G is associated with cognitive alterations and visual disturbances. Microgravity exposure causes the shift of fluid to the upper part of the body, inducing an increase of intracellular and interstitial fluid volume [9,10]

Further experiments demonstrated the possibility to obtain a reliable JVP by a non-operator dependent plethysmography sensor compared with US methodology, which were both synchronized with the ECG [11].

It is well known how physical exercise induces favorable cardiovascular adaptation. In particular, aerobic exercise training leads to an improvement in maximal cardiac output which is the result of an enlargement in cardiac dimension, improved contractility, and increase in blood volume, allowing for greater filling of the ventricles and a consequent larger stroke volume. In addition, endurance training may also induce alterations in the vasodilator capacity, although such adaptations are more pronounced in individuals with reduced vascular function [12]. 

As regarding the effect of exercise on brain circulation most of the studies is related on the cerebral inflow and metabolism, [13] to the contrary the impact of exercise on the cerebral venous drainage is still not enough investigated.

Moreover, the scientific literature clearly shows that dynamic supine exercise modifies the cerebral venous outflow, in relation with arterial inflow changes in both anterior and posterior cerebral circulation [14]. 

There are limited data regarding the response of JVP to exercise because of its complex and/or invasive assessment. Recently, Kasai et al. simplified the JVP evaluation visually considering as high JVP the internal jugular venous pulsation on the right side above the right clavicle in the sitting position. In conclusion, higher JVP was associated with exercise intolerance and poor prognosis [15]. 

In this work, we hypothesized that the new wearable plethysmograph could be a useful tool to monitor exercise protocols in addition to the beat per minute (BPM) information. To this aim, the present study has been designed to test the feasibility of plethysmograph JVP to monitor different acute exercise condition.

## 2. Materials and Methods

Twenty (20) healthy volunteers among students and residents (13M-7F, mean age 25 ± 3 years, BMI: 21 ± 2 kg m^−2^) were consecutively included in the study.

All the subjects were defined physically active since they performed regular physical activity at least 4–5 times per week but not sport professional. All the subjects underwent cerebral venous and arterial US evaluation to exclude supra-aortic vessels impairments. Inclusion and exclusion criteria for the proposed study are listed below.

Inclusion criteria were:
Age from 18 to 30 years.BMI < 28 kg·m^−2^.Good health condition.

Exclusion criteria were:
Use of supplement affecting venous volume.Postural defects.Sport professionalism.

### 2.1. Experimental Activity

All the subjects underwent cervical strain-gauge plethysmography while performing randomly the following activities separated by a one-week period. 

Supine for 2 min: laying down on a bed, breathing normally.Standing still for 2 min: standing distributing the body load equally on the 2 feet, breathing normally.Leg-press 6-repetition maximum (6-RM): the 6-RM test techniques was performed according to the recommendation from the American College of Sports Medicine (ACMS) guidelines for exercise testing and prescription [16,17,18]. The subject warmed up, completing a number of submaximal repetitions. The 6-RM load was determined selecting an initial weight that is within the ~50–70% individual perceived capacity. Load is progressively increased by 5–10% until the subject was able to complete the 6th repetition but not over. Once the ideal load was identified, the subject was invited to perform 3 sets of 6-RM with and interval of 30 s in between the series.2 km walking test: the test followed the description made by Oja et al. [19]. Initially, a 5-min warm-up was performed, with the aim to achieve the maximum comfortable speed on treadmill that the participants were able to maintain for 2 km, by rating of perceived exertion 13–14 (Borg 0–20) [20]. The duration of the test varied from a minimum of 17 to a maximum of 21.5 min. Two experienced exercise physiologists supervised all the exercise tests and sessions.

The focus of our analysis was on *a* and *x* peaks of the JVP curve since they reflect the volume of cardiac filling. To this aim we introduced the parameter (Δ*ax*), which is the amplitude difference between *a* and *x* peaks. 

All the tests were performed in the same room with a temperature set between 23–25 °C, between 9 and 11 A.M. During the different exercises, the subjects were requested to wear sport clothes, tennis shoes, and to maintain a good level of hydration drinking 500 mL of water 30 min before starting the tests [21].

### 2.2. Wearable Cervical Plethysmography System

The wearable cervical plethysmography system (Figure 2) includes a necklace-shaped strain gauge sensor, electrodes for conventional 3-lead ECG analysis and a portable electronic unit (PEU). Strain gauge sensors can be made with resistive or capacitive elements and are capable of measuring both deformation and forces [22,23].

The sensing element of the plethysmography system for monitoring the JVP waveform is the capacitive strain gauge sensor developed by LEAP technology ApS, Aabenraa, Denmark. It is a dielectric elastomer, which comprises an insulating film sandwiched between two deformable electrodes, thus forming a stretchable capacitor. Being capacitive makes the sensor extremely accurate, so when a minimal deformation occurs, the change in capacitance value is measured. Its high elasticity and mechanical robustness make the sensor ideal for a wearable solution because it can be placed on the skin surface without affecting the way people move. As is clearly visible in Figure 2a, the capacitive strain gauge sensor is formed by a stretchable sensitive zone, and two non-stretchable areas at its ends, covered with Velcro, to wear the sensor around the neck. The stretchable zone is 200 mm length. The strain gauge sensor was placed in anterior part of the neck at the level of the IJV, adherent to the skin. To avoid eventual detection errors, the sensor was placed in the lower part of the neck, closest to the right atrium, allowing properly the JVP waveform detection and excluding as much as possible artifacts.

The placement of electrodes for the 3-lead ECG is shown in Figure 2b. Particularly, the RA electrode is placed under right clavicle near right shoulder within the rib cage frame, the LA electrode is placed under left clavicle near left shoulder within the rib cage, and the LL electrode is placed on the left side below pectoral muscles lower edge of left rib cage.

For the characterization of both strain gauge and ECG sensor, please refer to [11].

About the PEU, which is shown in Figure 2c, it contains the circuitry to synchronously acquire both the JVP and ECG signals at a sampling rate of approximately 66 Hz. All the data are saved to text files for post-processing purposes. In addition, the PEU enables real-time data display on a PC through a Bluetooth wireless communication interface.

To assess the reproducibility of the signal acquired with the cervical plethysmography system for all subjects, a visual check of the JVP waveform on the PC screen was performed for a period of about 5 min before starting the tests. No changes were found in the offset and shape of the JVP, except for physiological variation in the signal related to swallowing.

### 2.3. Data Processing

Analysis of acquired data was carried out with MATLAB^®^ software version R2022b. Both JVP and ECG signals were filtered to remove the noise in order to highlight the cardiac oscillation. The DC component on signals, as well as the breathing contribution were eliminated through a high-pass filter with cut-off frequency of 0.5 Hz for both the JVP and ECG signals. Then, to eliminate the high-frequency components of noise, a low-pass filter with cut-off frequency of 4.5 Hz was used for the JVP signal, while a low-pass filter with cut-off frequency of 15 Hz was used for the ECG [11]. 

To compare the JVP signals acquired while carrying out the different exercise, it was decided to make a normalization between data. Considering the JVP signal acquired when users were in supine position as a reference, it was calculated the maximum a value ($a_{M,\sup}$) for each subject, and then all the JVP waveform were normalized with respect to the calculated $a_{M,\sup}$ value. This procedure was repeated for each individual subject. Once we got the normalized JVP signals, an automatic algorithm was developed to find the *a* and *x* values, as well as to calculate the difference Δ*ax* for each cardiac cycle on all the JVP waveforms. The algorithm was also designed to calculate the BPM on the ECG signals.

### 2.4. Statistical Analysis

The statistical analysis was performed using Prism 8, vers.8.2.1, 2019 (GraphPad Software Inc., San Diego, CA, USA). The data were expressed as mean and standard deviation values. Kolmogorov-Smirnov test was used to assess the data distribution. Non-parametric Friedman test was used to calculate the differences among *a*, *x*, and Δ*ax* values and heartbeat rate, measured during the supine, standing, and acute aerobic or strength exercise. 

The significance level was set at *p*-value < 0.05.

## 3. Results

For each subject, JVP and ECG signals were acquired for the diverse activities: lying supine, standing, leg press and walking. Totally, 80 files were saved in .txt format and, for each file, appropriate filtering and detection of a and x peaks on the normalized JVP waveform were carried out. Moreover, the calculation of the BPM values was obtained for all the ECG acquired signals.

Figure 3 shows an example of data acquisition for a subject lying on a bed.

In Figure 3a, the subject is visible wearing the plethysmography system. The PEU collects the signals and send them wirelessly to a PC for real-time data visualization, Figure 3b. The synchronized, filtered JVP and ECG signals are then shown in Figure 3c, where the peaks of interest on the JVP waveform are highlighted in red.

To display the distribution of results on the exercises carried out by the recruited subjects, Figure 4 shows the boxplots grouping the calculated values for the Δax parameter normalized with respect to the $a_{M,\sup}$ value, as shown in Figure 4a, and the heartbeat rate in BPM, in Figure 4b.

Table 1 summarizes the results shown in Figure 4. Mean and standard deviation (σ) values, as well as the results of the statistical analysis are listed in Table 1.

Furthermore, an analysis of the distribution of the Δ*ax* parameter over the time was carried out for all the activities to assess if the amplitude of this parameter changes during the exercise. To this purpose, a time window of few seconds was selected and, on each interval, the mean value of the normalized Δ*ax* parameter was calculated. Then, a linear regression using the least squares method was applied to find the line of best fit for the obtained data over the time (*t*). Among the four considered activities, only during walking we found a correlation between the normalized Δ*ax* and *t*. In particular, focusing our analysis on the central part of the exercise, we noticed the aforementioned correlation in the 45% of the subjects and, even if rather wide oscillation around the best regression line were present, a general increasing trend of Δ*ax* over the time was observed in these cases. To reduce the oscillation amplitude, we selected a time window of 10 s. The choice of a simple linear regression was due to avoid overfitting of the data. Furthermore, a liner function was sufficient to catch the trend we are interested in. The range of R^2^ and of the slope of the regression line turned out to be wide, with the mean value ± the standard deviation of these parameters being 0.45 ± 0.18 and 0.006 ± 0.005 s^−1^, respectively. In Figure 5, the variation of the normalized Δ*ax* with time the is depicted for one the subjects who showed correlation among these variables during walking. The regression equation and the determination coefficient are reported in the caption of the figure.

## 4. Discussion

In this study, a wireless, non-invasive, wearable cervical system has been used to synchronously acquire the JVP and ECG signals while carrying out diverse exercises. The use of the proposed system can represent an innovation because such measurements are usually carried out with an ultrasound machine. However, an ultrasound examination requires the presence of a healthcare operator, whereas using the wearable system the user can independently monitor his or her own health thanks to the device’s ease of use [3,24]. 

Another useful technique for acquiring information about blood flow is the photoplethysmography, which is sensitive to the pulsation component of blood, usually arterial [25,26], but which has also recently been used to describe venous pulsation at the jugular level [6].

The feasibility of JVP monitoring in the proposed postural and exercise condition was particularly encouraging because it paves the way to further application in order to extract non-invasively and on the field important cardiovascular information. Moreover, we evaluated the consistency of the obtained measurements. 

Looking at Figure 3, the JVP and ECG traces appear not synchronous with respect to the physiologic correspondence between the JVP and ECG waveforms [27,28,29], which considers the point for the JVP measurement directly in the right atrium. In the present study, the acquisition of the JVP waveform was carried out by using the cervical plethysmography system at the level of the neck, at an anatomical point spatially located cranially respect to the right atrium. For this reason, a time interval exists between the maximum value of the JVP waveform, (i.e., a peak) when acquired in two different anatomical positions. Since the jugular is more cranial with respect to the right atrium, the a peak measured with the cervical plethysmography system will be located later on the temporal axis than the same value for the a peak measured with the central venous catheter [30].

Figure 4a shows us that cardiac filling, expressed by differences between peak *a* and *x*, decreases when the subjects change the posture from supine to standing. This simple variation means that our device is able to detect the redistribution of blood below the diaphragmatic line occurring standing position [31,32].

In addition, physiology tells us that the jugular cross-sectional area (CSA) in upright is significantly reduced. In upright, the jugular intravenous pressure becomes negative due to the vein location above the heart. Consequently, blood is shunted into the vertebral venous plexus as an alternative venous pathway [33].

Thus, it is reasonable to conclude that the data herein presented are consistent with human physiology. 

The more innovative and interesting part of our research protocol is, of course, linked to the monitoring of JVP under exercise condition. By comparing our findings during the leg press exercise and walking exercises, the calculated value of the Δ*ax* significantly increases respect to static conditions. The Δ*ax* value for the leg press was approximately three times higher, while it was nine times higher during walking. 

Regarding the analysis on the distribution of the ∆*ax* parameter over the time, the resulting linear regression equations show that the distribution of the Δ*ax* parameter for the subject in supine and standing position, as well as while doing the leg press is almost constant over the time. Conversely, during walking, there is an increase of the value of the Δ*ax* parameter over the time (Figure 5). Therefore, both the exercise load and duration seem to impact on the Δ*ax* parameter.

In addition to cardiovascular function, the Δ*ax* parameter could be also useful for the assessment of cerebral drainage. Indeed, in challenging condition such as spaceflight, Δax showed a microG time-exposure relation, decreasing during flight permanence, followed by a normalization at return to the Earth [8,9,10]. Therefore, the use of this parameter becomes useful for monitoring the influence of exercise on cerebral venous drainage.

Regarding the technological advancement, the current wearables include energy harvesting solutions to increase their operating time, and machine learning models that can automatically detect and predict diseases, from cardiovascular to neurodegenerative [34,35,36,37]. A further practical aspect can result from the use of wearables for monitoring exercise programs and personalized care at distance, contributing to the spreading of telemedicine [38,39,40,41].

In the present work, the wearable cervical plethysmography system allows to wirelessly transmit data on a PC for real-time visualization, and the developed algorithm is able to filter the noise and to detect the peak of interest on the JVP waveforms automatically.

Here, in the proposed work, a novel system able to measure the JVP waveform is presented but the obtained information refer to the elongation of the strain-gauge sensor. Our future perspective regards the validation of the cervical plethysmography system in the clinical practice to obtain precise CVP values. 

Some limitations need to be discussed regarding the measurement obtained by the novel device. The modification of the neck volume can be also determined by voluntary muscular contrition, especially dynamic condition. For this reason, we strongly recommended to the subject under investigation should not contract the neck muscles during the experimental session. Furthermore, we also recommended during the walking to hold the treadmill hands support in order to avoid undesirable arm oscillation. These are, of course, important limitations in the development of future use of the novel plethysmography system aimed to assess the JVP under exercise condition. If from one side our investigation seems to support a future implication in this field, a technological improvement will be necessary on the other. In addition, the limited number of included subjects is necessarily to be reported among the study shortcomings. However, the purpose of this study was to verify the feasibility of the JVP plethysmographic assessment during dynamic conditions. Certainly, it will be fundamental to expand the data collection to a greater number of subjects as well as include different populations (not only heathy subject but also patients) and various types of exercise in order to verify the JVP trend.

Finally, by considering the limitation above, we avoided in this study to use exercises requiring maximal effort which are potentially more interesting to assess the cardiovascular adaptation. 

## 5. Conclusions

The exercise monitoring by means of the novel JVP plethysmography system is feasible using submaximal exercise, and it provides an additional parameter to the heartbeat rate parameter. The wearable, cost-effective device presented herein can provide important parameters for monitoring both cardiovascular and cerebral venous systems, even during exercise. For the rehabilitation and adapted physical exercise experts, it will also be of fundamental importance for adjusting the rehab and exercise programs at a distance, using telemedicine. Further technological advancement of the sensor, aiming to reduce motion artifacts, is required for a widespread use of non-invasive JVP for exercise monitoring.

## Figures and Tables

**Figure 1 diagnostics-12-02407-f001:**
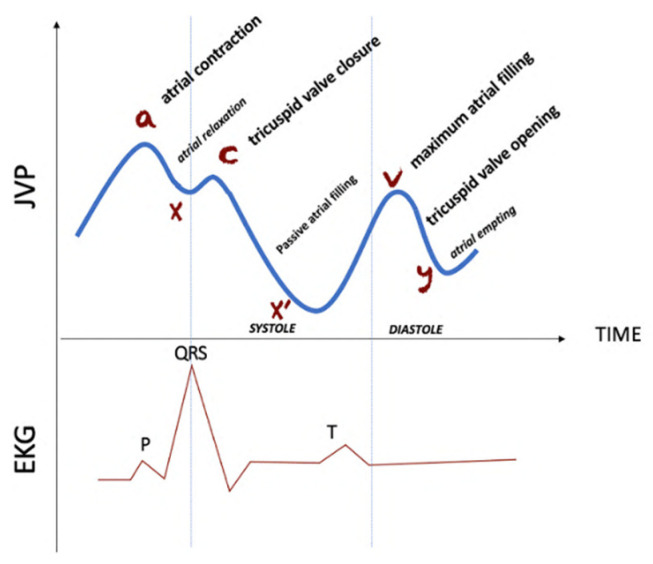
JVP waveform synchronized with ECG.

**Figure 2 diagnostics-12-02407-f002:**
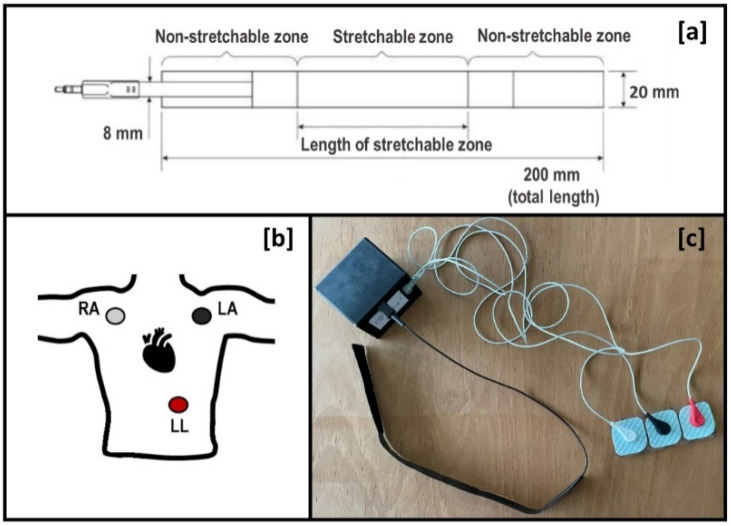
The plethysmography system: capacitive strain gauge sensor (**a**), electrode placement for 3-lead ECG (**b**), portable electronic unit (**c**).

**Figure 3 diagnostics-12-02407-f003:**
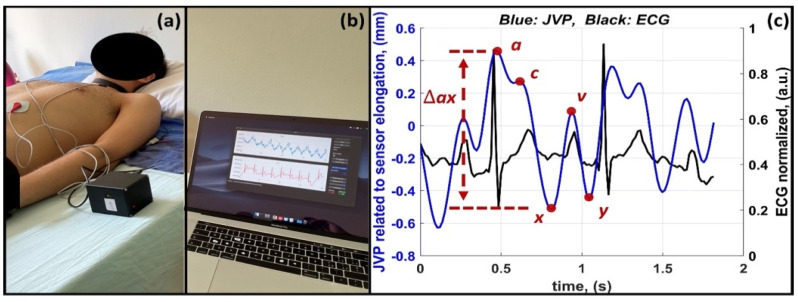
Synchronized acquisition of JVP and ECG signals. A subject lying on a bed (**a**); real-time data visualization (**b**); post-processed JVP and ECG signals (**c**).

**Figure 4 diagnostics-12-02407-f004:**
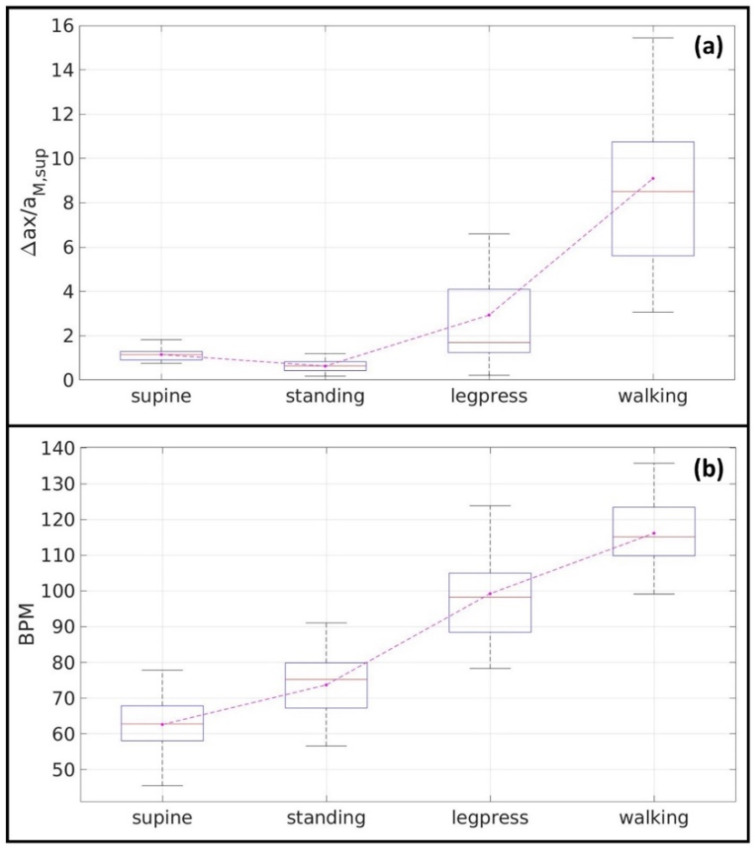
The Δ*ax* parameter normalized with respect to the $a_{M,\sup}$ value (**a**), and the heartbeat values (**b**).

**Figure 5 diagnostics-12-02407-f005:**
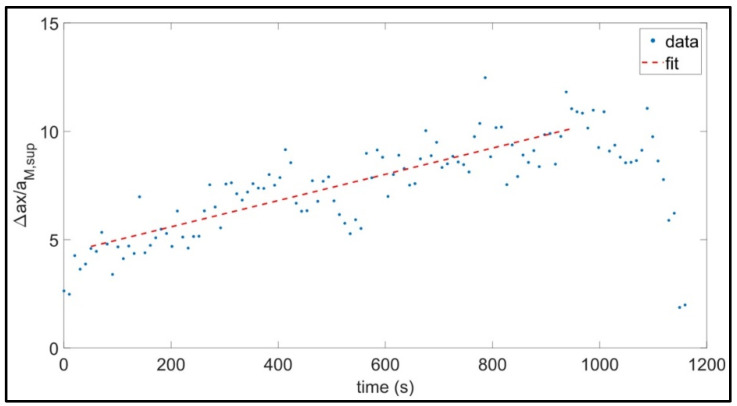
Example of correlation between the normalized ∆*ax* parameter and time for an individual subject during walking. The regression equation is Δ_*ax*/*a*M,sup_ = 4.379 + 0.006 t, for t in the range (50, 950) s, and the determination coefficient is R^2^ = 0.70.

**Table 1 diagnostics-12-02407-t001:** The dimensionless normalized values obtained for *a*, *x* and ∆*ax* parameter, as well as the heartbeat rate calculated during static (supine and standing) and dynamic (leg press and walking) exercises.

JVP Parameter	Supine: Mean ± σ	Standing: Mean ± σ	Leg Press: Mean ± σ	Walking: Mean ± σ	*p*-Value:
$a:\;\left(\frac{\bar{a}}{a_{M,sup}}\right)$	0.5 ± 0.1	0.3 ± 0.1	1.3 ± 1.2	4.7 ± 2.8	0.0001
$x:\;\left(\frac{\bar{x}}{a_{M,sup}}\right)$	−0.6 ± 0.2	−0.3±0.1	−1.6 ± 1.5	−4.3 ± 2.3	0.0001
$\Delta ax:\;\left(\frac{\bar{\Delta ax}}{a_{M,sup}}\right)$	1.1 ± 0.3	0.6± 0.2	2.9 ± 2.6	9.1 ± 5.1	0.0001
heartbeat rate (BPM)	63 ± 9	74 ± 10	99 ± 12	116 ± 10	0.0001

*p* level was calculated by means of non-parametric Friedman Test.

## Data Availability

The data that support the findings of this study are available from the corresponding author, [A.P. (Antonino Proto)], upon reasonable request.

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
