# Peer review of "The Effect of Submaximal Exercise on Jugular Venous Pulse Assessed by a Wearable Cervical Plethysmography System"

_diagnostics, 2022, doi:10.3390/diagnostics12102407_

Round 1
Reviewer 1 Report
Did you test JVC two or three times in the same person to check reproducibility?Due to the limitations you already outlined (neck position, non strenuous exercise etc)How could this measurement applied in clinical practice during movement? Did you check reproducible results at rest in diseased people? How hydratation or poor hydtatation can interfere?
Author Response
Response to Reviewer 1 Comments
We deeply thank you the Reviewer for his/her time and effort. We reviewed the manuscript according to the comments and we reported the detailed answers.
- Did you test JVP two or three times in the same person to check reproducibility?
Response: we agree. In the revised version of the manuscript, we added the following sentences at the end of the Paragraph 2.2 (Page 4 line 167): “To assess the reproducibility of the signal acquired with the cervical plethysmography system for all subjects, a visual check of the JVP waveform on the PC screen was performed for a period of about 5 minutes before starting the tests. No changes were found in the offset and shape of the JVP, except for physiological variation in the signal related to swallowing.”
- Due to the limitations you already outlined (neck position, non strenuous exercise etc), how could this measurement applied in clinical practice during movement?
Response: we thank the Reviewer for this comment, which drive us to implement our conclusion including future perspectives of our research. We strongly believe that the wearable, cost-effectiveness device can provide important parameters for monitoring both cardiovascular and cerebral venous systems, even during exercise. For the rehabilitation and adapted physical exercise experts, it will be of fundamental importance also for adjusting the rehab and exercise programs at distance, using telemedicine. Moreover, in our Lab we are studying different sensors aimed to reduce as much as possible swallowing and respiratory artifacts.
Page 10 line 358 “The herein presented wearable, cost-effectiveness device can provide important parameters for monitoring both cardiovascular and cerebral venous systems, even during exercise. For the rehabilitation and adapted physical exercise experts, it will be of fundamental importance also for adjusting the rehab and exercise programs at distance, using telemedicine. Further technological advancement on the sensor, aimed to reduce motion artifacts, is required for a widespread use of non-invasive JVP for exercise monitoring.”
- Did you check reproducible results at rest in diseased people?
Response: So far, we did not test the sensor on diseases subjects. It will be one of our main goals for future researches.
- How hydration or poor hydration can interfere?
Response: The hydration level can of course interfere with the filling of the IJV and in turn with the related crossectional area, therefore for the standardization of the protocol we decided to force all the subjects to drink 500 ml of water.
We now added a new reference n° 21 in which it has been reported this concept: “Pelizzari L, Laganà MM, Jakimovski D, Bergsland N, Hagemeier J, Baselli G, Zivadinov R. Neck Vessel Cross-Sectional Area Measured with MRI: Scan-Rescan Reproducibility for Longitudinal Evaluations. J Neuroimaging. 2018;28:48-56.

Reviewer 2 Report
This paper describes a method for jugular venous pulse (JVP) monitoring by plethysmography sensor and shows the results obtained from the cohort of 20 young individuals in resting and after submaximal exercise. The topic of the article and the proposed method are very interesting. Overall, the paper is well organized and reports an interesting application of wearable instrumentation for cardiovascular diagnosis.
However, some points need to be improved. Please consider the following comments and recommendations:
- In the Abstract, please do not be so drastic: “The jugular venous pulse (JVP) is a crucial parameter of efficient cardiovascular function”( line 16). It will be better to use “… one of the crucial parameters…” because there are others, not less important parameters to evaluate cardiovascular functions.
- Introduction: generally well described, except in the 47 – 54 lines and Fig 1, which I consider incorrect. The nature of JVP shall be explained from the physiological features with the typical ECG signals for cardiac time reference purposes: the waves a, c and v, the descents x (actually x1+x2) and y. I do not understand why in your Fig.1 a wave (atrial systole) is coincident with QRS and not the P wave, as it shall and as it is reported in the literature ( for example, in your ref.6). It is a crucial point for the article because later you represent an experimental picture in the Fig.3c)! You can use the terms “peak” for your signal processing end call peak x and y, but not for the JVP physiology explanation.
Another point: the image of Fig 1 needs to be improved in quality of the image, as well as the reference is missed, if it is not of your authorship.
-In the Materials and Methods it is important to clarify the following:
· Why the cohort is so small, only 20 persons? This is a noninvasive test experiment so, at least, 30 persons shall be considered;
· Why do the inclusion criteria consider only subjects with BMI < 28? For the feasibility study, it will be interesting to see if the plethysmography sensor works well in obese subjects.
· Please clarify how the strain gauge sensor was placed in order to correctly access the right internal jugular vein and how to avoid the right external jugular vein or carotid artery pulsations.
- Results shall be very much improved: Please consider the following:
· In Table 1 (line 217) are not “ values of a and x peaks” as it is written, but normalized adimensional values after statistical treatment.
· It will be important to represent the absolute values for a, c and v waves and x deflection for all 20 subjects, at least for one of two different conditions: resting (lying, supine or standing) and after submaximal exercise ( leg press or
· Since the JVP represent the important findings in the Central Venous Pressure (CVP) it is important to proceed with calibration of your plethysmography sensor elongation with the pressure and report the results. Please, explain this issue.
· Regarding the results of the fitting and presented equations for a correlation with the time of exposure to exercise (lines 224 – 233) I consider the linear regression wrong. Just from a little look at the distribution of the Δax parameter over the time during walking presented in Figure 5, it seems that non-linear fitting is better, at least a 2nd degree polynomial equation. In any case, the R-value for the regression mandatory shall be reported.
- Discussion section shall be better organised and improved regarding the considerations above mentioned.
· Please explain the overlapping of JVP and ECG traces reflects the physiologic correspondence between the JVP peaks and the ECG waveform (lines 251-253)
· The statement in lines 243-246 is scientifically incorrect and shall be modified or waived: the photoplethysmography (PPG) main application is not to measure the arterial pulse wave propagation. PPG is sensitive to the pulsation component of blood (usually arterial), and the experimental set of PPG is not more complex or specific than the described plethysmography sensor.
· In addition, it will be interesting to see analyses and discussions of JVP (the Δax parameter) variation with cardiac output.
· The results analysis would benefit from a deeper comparison with results obtained by other authors on the same JVP problem.

Author Response
Response to Reviewer 2 Comments
This paper describes a method for jugular venous pulse (JVP) monitoring by plethysmography sensor and shows the results obtained from the cohort of 20 young individuals in resting and after submaximal exercise. The topic of the article and the proposed method are very interesting. Overall, the paper is well organized and reports an interesting application of wearable instrumentation for cardiovascular diagnosis.
However, some points need to be improved. Please consider the following comments and recommendations:
We deeply thank the Reviewer for her/his time, effort, and valuable feedback on our submission. We have now replied to all the points and made the related changes in the manuscript.
-In the Abstract and Introduction, it is important to clarify the following:
- In the Abstract, please do not be so drastic: “The jugular venous pulse (JVP) is a crucial parameter of efficient cardiovascular function”( line 16). It will be better to use “… one of the crucial parameters…” because there are others, not less important parameters to evaluate cardiovascular functions.
Response: We agree with the Reviewer, and now we changed the abstract according to the suggestion. Page 1 Line 16 “The jugular venous pulse (JVP) is a one of the crucial parameters of efficient cardiovascular function”.
- In Introduction: generally well described, except in the 47 – 54 lines and Fig 1, which I consider incorrect. The nature of JVP shall be explained from the physiological features with the typical ECG signals for cardiac time reference purposes: the waves a, c and v, the descents x (actually x1+x2) and y. I do not understand why in your Fig.1 a wave (atrial systole) is coincident with QRS and not the P wave, as it shall and as it is reported in the literature ( for example, in your ref.6). It is a crucial point for the article because later you represent an experimental picture in the Fig.3c)! You can use the terms “peak” for your signal processing end call peak x and y, but not for the JVP physiology explanation.
Response: We apologize for the figure 1, that evidently appears out of phase between JVP and ECG. We agreed that the P wave of the ECG corresponds to the A wave of the JVP. We now changed it accordingly in the “New Figure 1”.
Moreover, we corrected the terms “peak” in the JVP, implementing as suggested all the physiology explanation at Page 2 Line 47: “The JVP waveform is composed by three ascents and three descents waives, which correspond to the pressure variation of the cardiac phases: atrial contraction (wave a) synchronized with P peak of ECG, followed by atrial relaxation (wave x) and tricuspid valve closure (wave c). After the QRS complex of the ECG the ventricular systole starts while passive atrial filling occurs followed by pressure drop (wave x’). Subsequently, after the T peak of ECG corresponding to ventricular repolarization, the maximum atrial filling will be obtained (wave v) before the tricuspid valve opening. This latter causes a sudden pressure decreases coincident with ventricular filling (wave y), and the cycle will start again.”
- Another point: the image of Fig 1 needs to be improved in quality of the image, as well as the reference is missed, if it is not of your authorship.
Response: we now improved the quality of the New Figure 1 (600dpi), moreover we confirm that no reference is needed because it is of our authorship.
-In the Materials and Methods, it is important to clarify the following:
- Why the cohort is so small, only 20 persons? This is a noninvasive test experiment so, at least, 30 persons shall be considered.
Response: We agree with the Reviewer, that 20 people is a small cohort and now we reported it in the discussion among the limitations of the study. Page 9 line 346” In addition, the limited number of included subjects is necessarily to be reported among the study shortcomings. However, the purpose of this study was to verify the feasibility of the JVP plethysmographic assessment during dynamic conditions in healthy subjects. Certainly, it will be fundamental to expand the data collection to a greater number of subjects as well as include different populations (not only heathy subject but also patients) and various types of exercise in order to verify the JVP trend.”
Moreover, following the comment of the Reviewer, we now reanalyzed our statistics using the non-parametric Friedman Test which is more appropriate for less than 30 cases (which is considered a cut-off for the application of parametric statistics). The significance remains the same, therefore we now reported the changes in both in the statistical analysis and as a footnote of the table 1.
Page 5 line 148 ”Non-paramteric Friedman Test was used to calculate the differences among a, x, and Δax values and heartbeat rate, measured during the supine, standing and acute aerobic or strength exercise.”
We strongly agree with the Reviewer that an accurate sample size calculation can tell us how many subjects need to be included, the present study provide us significant outcome on which an independent statistician can calculate the sample size.
- Why do the inclusion criteria consider only subjects with BMI < 28? For the feasibility study, it will be interesting to see if the plethysmography sensor works well in obese subjects.
Response: We agree with the Reviewer observation, it will be more than interest to test the sensor also in obese subjects and this point is among our purposes for the future investigation.
The main goal of this study was to investigate healthy subject which were able to perform any kind of exercise, therefore we prefer to avoid any pathological cases. From clinical point of view a person with BMI over 28 has to be consider among the pathological cases. Moreover, the obese condition can interfere also with exercise performance so prefer to plan an appropriate study, with specific exercise test for pathological cases.
- Please clarify how the strain gauge sensor was placed in order to correctly access the right internal jugular vein and how to avoid the right external jugular vein or carotid artery pulsations.
Response: We now added in the methods section the procedure to proper position the sensor in order to avoid as much as possible interference from other vessels. Page 4 line 152 “The strain gauge sensor was positioned in anterior part of the neck at the level of the IJV, adherent to the skin. To avoid eventual detection errors, the sensor was placed in the lower part of the neck, closest to the right atrium, allowing properly the JVP waveform detection and excluding as much as possible artifacts.”
Regarding the carotid pulse, it corresponds to the wave C generated by the closure of the tricuspid and was not considered in our analysis. The external jugular vein has little pulsation due to the outlet in the subclavian vein, moreover it runs in the lateral part of the neck, so it doesn’t interfere with our sensor who was placed anteriorly as we now reported in our description.
- Results shall be very much improved: Please consider the following:
- In Table 1 (line 217) are not “values of a and x peaks” as it is written, but normalized adimensional values after statistical treatment.
Response: We agree. In the new version of the manuscript, the caption of Table 1 has been changed, as it follows, Page 7 line 229: “The dimensionless normalized values obtained for a, x and ∆ax parameter, as well as the heartbeat rate calculated during static (supine and standing) and dynamic (leg press and walking) exercises”.
- It will be important to represent the absolute values for a, c and v waves and x deflection for all 20 subjects, at least for one of two different conditions: resting (lying, supine or standing) and after submaximal exercise ( leg press or)
Response: We agree in general terms that an absolute value is more straightforwardly interpretable. However, the obtained absolute values were within a wide range due to the large variability of the neck circumferences of subjects.
For this reason, we decided to normalize the data with respect to the maximum value of the signal acquired in supine position.
- Since the JVP represent the important findings in the Central Venous Pressure (CVP) it is important to proceed with calibration of your plethysmography sensor elongation with the pressure and report the results. Please, explain this issue
Response: CVP, as correctly observed by the Reviewer, is certainly correlated with the JVP since it is actually the measurement to express the CVP itself. However, it is an invasive parameter and thus is not suitable for the purpose of pletismography calibration. Furthermore, ultrasound venous pulse may permit a reliable estimation of CVP (Ref. n. 7).
In Literature exist many validated ways to obtain CVP values starting from the information acquired with the ultrasound machine on the blood velocity and vein cross-sectional area.
We now discussed it at Page 9 Line 333 “Here, in the proposed work a novel system able to measure the JVP waveform is presented but the information that we get are related to the elongation of the strain-gauge sensor. Our future perspective regards the validation of the cervical plethysmography system in the clinical practice to obtain precise CVP values.”
- Regarding the results of the fitting and presented equations for a correlation with the time of exposure to exercise (lines 224 – 233) I consider the linear regression wrong. Just from a little look at the distribution of the Δax parameter over the time during walking presented in Figure 5, it seems that non-linear fitting is better, at least a 2nd degree polynomial equation. In any case, the R-value for the regression mandatory shall be reported.
Response: Taking a cue from the reviewer’s comment, we have investigated further and modified this part of the manuscript, justifying the choice of a linear regression and reporting the determination coefficient (R2). In particular, the Lines 221-233 of the old version of the manuscript have been rephrased as follows Page 7 line 245:
“Furthermore, an analysis on the distribution of the Δax parameter over the time was carried out for all the activities to assess if the amplitude of this parameter changes during the exercise. To this purpose, a time window of few seconds was selected and, on each interval, the mean value of the normalized Δax parameter was calculated. Then, a linear regression using the least squares method was applied to find the line of best fit for the obtained data over the time (t). Among the four considered activities, only during walking we found a correlation between the normalized Δax and t. In particular, focusing our analysis on the central part of the exercise, we noticed the aforementioned correlation in the 45% of the subjects and, even if rather wide oscillation around the best regression line were present, a general increasing trend of Δax over the time was observed in these cases. To reduce the oscillation amplitude, we selected a time window of 10 s. The choice of a simple linear regression was due to avoid overfitting of the data. Furthermore, a liner function was sufficient to catch the trend we are interested in. The range of R2 and of the slope of the regression line turned out to be wide, being the mean value ± the standard deviation of these parameters 0.45 ± 0.18 and 0.006 ± 0.005 s-1, respectively. In figure 5 is depicted the variation of the normalized Δax with time the for one the subjects who showed correlation among these variables during walking. The regression equation and the determination coefficient are reported in the caption of the figure.”
In the revised version of the manuscript, we reported an improved version of Figure 5.
- Discussion section shall be better organised and improved regarding the considerations above mentioned.
- Please explain the overlapping of JVP and ECG traces reflects the physiologic correspondence between the JVP peaks and the ECG waveform (lines 251-253)
Response: Following the Reviewer suggestion in the revised version of the manuscript, we rephrased the following sentences explaining in detail the phenomenon in the discussion session: Page 8 Line 287 "Looking at Figure 3, the JVP and ECG traces appear not synchronous with respect to the physiologic correspondence between the JVP and ECG waveforms, which considers the point for the JVP measurement directly on the right atrium. In the present study, the acquisition of the JVP waveform was carried out by using the cervical plethysmography system at the level of the neck, at an anatomical point spatially located cranially respect to the right atrium. For this reason, it exists a time interval between the maximum value of the JVP waveform, i.e., a peak, when acquired in two different anatomical positions. Since the jugular is more cranially respect to the right atrium, the a peak measured with the cervical plethysmography system will be located later on the temporal axis than the same value for the a peak measured with the central venous catheter” new ref 30 [Sisini, F. Physical description of the blood flow from the internal jugular vein to the right atrium of the heart: new ultrasound application perspectives. arXiv2016;1604.05171]”
- The statement in lines 243-246 is scientifically incorrect and shall be modified or waived: the photoplethysmography (PPG) main application is not to measure the arterial pulse wave propagation. PPG is sensitive to the pulsation component of blood (usually arterial), and the experimental set of PPG is not more complex or specific than the described plethysmography sensor.
Response: agreed. In the revised version of the manuscript the statement is now modified as it follows, Page 8 Line 279: “Another useful technique for acquiring information about blood flow is the photoplethysmography, which is sensitive to the pulsation component of blood, usually arterial [25,26], but which has also recently been used to describe venous pulsation at the jugular level [6].”
In addition, it will be interesting to see analyses and discussions of JVP (the Δax parameter) variation with cardiac output.
Response: we totally agree with the reviewer it will be interesting correlate cardiac output and Δax, in this preliminary phase we did not have the possibility to monitor cardiac output which requires ultrasound machine equipped with cardiac probe, but for sure it is among our future perspectives.
The results analysis would benefit from a deeper comparison with results obtained by other authors on the same JVP problem.
Response: to the best of our knowledge papers assessing the JVP non-invasively during exercise were not available.

Round 2
Reviewer 1 Report
There was an adequate revision process
Reviewer 2 Report
The corrections introduced by the authors are sufficient for the article to be accepted for publication.